# Low Temperature In Situ Synthesis of ZnO Nanoparticles from Electric Arc Furnace Dust (EAFD) Waste to Impart Antibacterial Properties on Natural Dye-Colored Batik Fabrics

**DOI:** 10.3390/polym15030746

**Published:** 2023-02-01

**Authors:** Istihanah Nurul Eskani, Edia Rahayuningsih, Widi Astuti, Bidhari Pidhatika

**Affiliations:** 1Center for Craft and Batik, Ministry of Industry, Jl. Kusumanegara No. 7, Yogyakarta 55166, Indonesia; 2Department of Chemical Engineering, Gadjah Mada University, Jl. Grafika No. 2, Yogyakarta 55281, Indonesia; 3Indonesia Natural Dye Institute (INDI), Jl. Kaliurang Km. 4 Sekip Utara, Kabupaten Sleman, Yogyakarta 55281, Indonesia; 4Research Center for Mining Technology, National Research and Innovation Agency (BRIN), Jl. Ir. Sutami Km. 15, Tanjung Bintang, Lampung Selatan 35361, Indonesia; 5Research Center for Polymer Technology, National Research and Innovation Agency, Republic of Indonesia—PRTPL BRIN Indonesia, Serpong, Tangerang Selatan 15314, Indonesia; 6Collaborative Research Center for Biomedical Scaffolds, National Research and Innovation Agency of the Republic Indonesia and Universitas Gadjah Mada, Jalan Denta No. 1, Sekip Utara, Yogyakarta 55281, Indonesia

**Keywords:** antibacterial activity, Batik fabric, CBD, in situ synthesis, natural dye, ZnO nanoparticles

## Abstract

Natural polymer (cellulose)-based fabric was colored using an environmentally friendly natural dye extracted from *Jalawe* (*Terminalia bellirica*) in the preparation of Batik fabric, a cultural heritage of Indonesia that is recognized by United Nations Educational, Scientific and Cultural Organization (UNESCO). Despite the significant favorable properties in terms of functions, environmental, and cultural aspects, the combination between natural polymer-based fabric and natural dyes makes the Batik fabric an ideal medium for bacterial growth, leading to lower product quality. In the quest for a sustainable, environmentally friendly, rich-in-culture, yet durable textile, this study aimed at the functionalization of natural dye (ND)-colored Batik fabric with antibacterial ZnO nanoparticles (ZnO NPs) synthesized from Electric Arc Furnace Dust (EAFD) waste. An in situ immobilization process with a Chemical Bath Deposition (CBD) method was explored at a pH range from 6 to 11 at 50 °C. Characterization methods include XRD, XRF, FESEM, EDX, FT-IR, tensile strength measurement, agar diffusion testing, and a CIE L*a*b* scale measurement. The XRD and XRF results showed that pure (>98%) ZnO NPs were formed at pH 11 of the CBD process. FESEM results demonstrated that the pure ZnO NPs either precipitated at the CBD reactor or were immobilized on the cellulose fabric, exhibiting distinct morphology compared to the non-pure ZnO NPs. EDX elemental analysis before and after washing demonstrated the durability of the ZnO NPs attachment, in which 84% of the ZnO NPs remained on the fabric after two washing cycles (equal to 10 cycles of home laundering). The FT-IR spectra provided information on the chemical functional groups, demonstrating the success of the ZnO NPs immobilization on the cellulose fabric through Van der Waals or coordination bonding. Moreover, the in situ immobilization of ZnO NPs enhanced the Batik fabric’s tensile strength but reduced its elongation. ZnO NP-functionalized Batik fabric that was treated at pH 10 and pH 11 showed antibacterial activity against *Staphylococcus aureus*. The CIE L*a*b* scale results showed that the immobilization process affects the color quality of the ND-colored Batik fabric. However, based on organoleptic observations, the color of the Batik fabric that was treated at pH 11 is still acceptable for *Jalawe* ND-colored Batik fabric.

## 1. Introduction

Cellulose is the most significant and readily accessible renewable natural polymer for textiles [1]. Importantly, when combined with motifs and colors, the natural polymer-based textiles (cotton) do not just serve functional purposes, but they frequently represent historical crafts that preserve the distinctive art and history of various cultures. One such readily recognizable example is Batik, a motif created by a selective dyeing technique on textile. Batik is an essential part of Indonesia’s intangible cultural heritage. It has been passed down the years and is frequently used in people’s daily lives. Batik motifs are estimated to come in over 3000 various colors and compositions, allowing for the fusion of indigenous culture with imported and creative design [2]. Natural dye (ND)-colored Batik fabric is favored by consumers [3] because it has an attractive appearance and is more environmentally friendly [4] compared to synthetic dye-colored fabric. Despite the significant aspects, fibers based on natural polymers provide an ideal surface medium for bacterial growth [5,6] because of their vast surface area and porous nature, especially in the presence of the right levels of nutrients, temperature, and moisture [7]. ND-colored cotton fabrics, which combine natural polymer fibers with natural dyes, provide such ideal medium and nutrients. Therefore, fabrics with attached natural dyes easily become dull and exhibit low color fastness [8]. In the quest for sustainable, environmentally friendly, rich-in-culture, yet durable textiles, especially in terms of color, antibacterial functionalization technology for ND-colored Batik fabric has been developed. Functional textiles have superior properties, such as anti-wrinkle, antibacterial, and anti-UV. The development of functional textiles can be achieved through the application of nanoparticles on textile materials [9]. 

Zinc oxide nanoparticles (ZnO NPs) are n-type semiconductors with a band gap energy of 3.37 eV. ZnO NPs have advantages over other nanoparticles in terms of cost-effectiveness, and excellent antibacterial and anti-UV properties. ZnO NPs have also been declared as a safe material by the US Food and Drug Administration (FDA) [10,11]. Due to their advantages, ZnO NPs are widely used in biological and biomedical applications, such as bioimaging, drug delivery, antidiabetic, anticancer, antifungal, antibacterial, and anti-inflammatory activities [12]. This study utilizes ZnO NPs to provide antibacterial properties to ND-colored Batik fabric. The immobilization of ZnO NPs on fabrics can be carried out by either in situ or ex situ methods. In the ex situ method, ZnO NPs in the form of a powder are suspended and then applied to the fabric; this can be performed by the pad-dry-cure, sonochemical, or plasma method. In the in situ method, the synthesis of ZnO NPs is directly carried out in the fabric fibers, so that the synthesis and application to the fabric are carried out in one process. The ex situ method has advantages; for example, among others, the size and shape of the ZnO NPs crystals are known so that the physical and chemical properties can be estimated. A disadvantage is that the adsorption capacity to the fabric is low and can easily agglomerate. The advantages of the in situ method include an even distribution of nanoparticles on the fabric, better adsorption and durability, and this method produces a strong bond between the nanoparticles and the fabric fibers [13]. The disadvantage of the in situ method is the difficulty in controlling the formation of the nanoparticles in the fabric fibers [14].

The disadvantage of ZnO NPs, in terms of textile functionalization, is that they do not have a good affinity with textile fibers. One effort to increase the affinity of ZnO NPs with fabric fibers is by using the in situ application process for applying ZnO NPs to fabrics [11]. The in situ process can be carried out using the Chemical Bath Deposition (CBD) method. CBD is a simple deposition method on the surface of a substrate by placing it in a vessel containing a supersaturated solution of metal salt precursors, complexing agents, and pH buffers [15]. The advantages of CBD include low cost, being able to be operated at low temperatures (<100 °C) and atmospheric pressure, and it is very simple and harmless [16,17]. Due to the advantages of the CBD method, it is considered suitable for nanoparticle applications on Batik cloth, especially ND-colored Batik.

In this study, Electric Arc Furnace Dust (EAFD), waste from the steel industry, was used as a raw material for the synthesis of ZnO NPs. The stainless steel industry is one of the priority industries to be developed in Indonesia during the 2020–2024 period [18]. Based on data from the Indonesian Iron and Steel Association (IISA), there are around 200 steel industries in Indonesia, mostly in West Java, Banten, East Java, and Sumatra. One of the industrial wastes is Electric Arc Furnace Dust (EAFD) [19]. EAF dust waste generated by the steel industry in Indonesia reaches a volume of 1680 m^3^/year [20]. The zinc content in this material can reach 50–60% so it has the potential to be reprocessed to produce zinc metals or compounds [21].

Research on antibacterial functionalization by imparting ZnO NPs onto fabric has been carried out by researchers. Javed et al. [22] succeeded in performing antibacterial, anti-UV and self-cleaning functionalization on cotton fabrics using the ultrasonically assisted in situ synthesis of ZnO NPs on cotton fabric at 90 °C and then curing at 130 °C. Tania and Ali [23] reported that the antibacterial properties of cotton fabric, along with better mechanical properties, were obtained by applying ZnO NPs combined with a polyethylene wax emulsion and binder onto fabric using padding, then drying at 90 °C and curing at 150 °C. The antibacterial functionalization of cotton fabrics using a deep coating method at room temperature was reported by Pintaric et al. [24]. Research on the antibacterial functionalization of Batik fabric has been carried out by the authors in previous studies, both ex situ [25] and in situ [26]. The results from both studies showed that the application of ZnO NPs can provide antibacterial properties to synthetic dye (SD)-colored Batik fabric. The current study embarks on applying ZnO NPs to ND-colored Batik fabrics using the in situ method. Compared to existing literature, there are several new aspects that are addressed in the current study, namely: (1) the use of EAF dust as the source of ZnO NPs; (2) the use of low temperature (<100 °C) during the in situ process; and (3) the use of ND-colored fabric as the substrate for the in situ immobilization of ZnO NPs. The combination of the above-mentioned materials and synthesis conditions has, to the best of the authors’ knowledge, not been found in the literature. 

## 2. Materials and Methods

### 2.1. Materials

Materials used for the leaching process were EAFD (PT Central Fortuna Steel, Bogor, Indonesia), HNO_3_ (Merck KGaA, Darmstadt, Germany), filter paper (Whatman No. 42, GE Healthcare, Buckinghamshire, UK), deionized water, and NaOH (Merck KGaA, Darmstadt, Germany). Materials for the preparation of the ND-colored Batik fabric were wax, cotton fabric, *Jalawe* (*Terminalia bellirica*) rind as a natural dye, non-ionic detergents, soda ash (Na_2_CO_3_), and alum mordant. The chemicals used for making the ND-colored Batik fabric were technical grade and purchased from a local market in Yogyakarta. 

### 2.2. Instrumentations

Inductive Coupled Plasma (ICP OES, Plasma Quant PQ 9100 Series, Analytik Jena US Inc., Upland, CA, USA) was used to determine the chemical composition of the leaching solution. The nanoparticles powder was characterized using X-ray Diffraction (XRD, X’pert 3 Powder, Malvern Panalytical, Almelo, The Netherlands) and X-ray Fluorescence (XRF, Epsilon 3 XLE, Malvern Panalytical, Almelo, The Netherlands) to determine its crystallinity and chemical composition, respectively. The Brunauer–Emmett–Teller (BET) surface area and Barret–Joyner–Halenda (BJH) pore size was analyzed using the nitrogen gas adsorption-desorption method at 77 K by a Micromeritics ASAP 2000 (Micromeritics Instrument Corp., Norcross, GA, USA). Field-Emission Scanning Electron Microscopy (FESEM EDX, Thermo Fisher Scientific Inc., Waltham, MA, USA) was used to observe the crystal shape of the ZnO NPs and the morphology of the ZnO NPs applied to the Batik fabric. The interaction between the ZnO NPs and ND-colored Batik fabrics was evaluated by Fourier Transform Infrared Spectroscopy (FT-IR, Shimadzu IRPrestige-21, Shimadzu, Kyoto, Japan) in the range 4000–400 cm^−1^. The durability of the ZnO NPs coating on ND-colored Batik fabric was investigated using a Launder-o-meter (Heraeus Deutschland GmbH & Co. KG, Hanau, Germany), then the morphology and chemical composition of the Batik fabrics before and after washing was evaluated using FESEM and Energy dispersive X-ray spectroscopy (EDX), respectively. The assessments of color differences used a colorimeter (AMT506, Tlead International Co., Ltd., Qingdao, China) and the mechanical properties of the fabric were tested using a Universal Testing Machine (UTM, J.T.M Technology Co., Ltd., Taichung, Taiwan). 

### 2.3. Methods

#### 2.3.1. The Leaching of EAFD

The leaching of EAFD was performed according to a previously published protocol [21]. First, an EAFD sieving was carried out to obtain a dust grain size of <75 µm. Then, the leaching process was carried out using 3 M nitric acid reagent at 80 °C for 5 h, with a 5 g/50 mL solid to liquid ratio (pulp density) and 200 rpm stirring speed. The leached solution was characterized using ICP to examine its chemical composition.

#### 2.3.2. Preparation of ND-Colored Batik Fabrics

White cotton fabrics were immersed in a non-ionic detergent solution (1 g/L) to clean the fabric from any contaminants and then they were dried. ND from *Jalawe* (*Terminalia bellirica*) rind was extracted following a previously published protocol [27] for 1 h, at 1:5 *w*/*w Jalawe* rind/water and at 100 °C. There were 3 (three) major steps in the preparation of the ND-colored Batik fabrics: (1) wax patterning onto white cotton fabric, (2) dyeing (coloring) process, and (3) wax removal. In the first step, a heated and melted wax was applied to the white cotton fabric using a Batik stamp (canting) following a desired pattern. In the second step, the wax-patterned fabric was dipped in *Jalawe* rind extract for approximately 15 min, then it was dried. The dip-and-dry cycle was repeated 5 times. Following the last coloring cycle, the fabric was immersed in 70 g/L alum mordant solution for 5 min to facilitate stronger bonding between the ND molecules and the cotton fibers. Rinsing with clean water and drying allowed for the removal of the ND and mordant residue in the second step. In the final (third) step, the wax was removed from the fabric by immersing the wax-patterned and colored fabric in boiling water. Soda ash (2 g/L) was added to the boiling water to accelerate the removal of the wax. The fabrics collected from the above-described steps were then called ND-colored Batik fabrics. 

#### 2.3.3. In Situ Immobilization of ZnO NPs on ND-Colored Batik Fabric Using a Chemical Bath Deposition (CBD) Method

The leaching solution was used as a precursor in the in situ ZnO immobilization onto Batik fabric. Firstly, NaOH (10%) was added dropwise to the leaching solution until it reached pH 5. This step was performed to precipitate unwanted metals (especially iron/Fe). The filtrate was then collected as a precursor solution. The 200 mL precursor solution was stirred and heated in a beaker. The Batik fabric sample was immersed into the stirred solution at 50 °C, followed by the dropwise addition of NaOH (1 M) until the pH solution reached a value of 6, 7, 8, 9, 10, or 11. During the addition of NaOH, the solution became milky-white, indicating the formation of Zn complexes and/or Zn(OH)_2_. After 10 min of heating, the ND-colored Batik fabric was removed from the solution, rinsed 2 (two) times using distilled water, and dried at room temperature. In the next step, the characterizations were performed in 2 (two) parts, i.e., (1) the characterization of ZnO NPs that were precipitated on the bottom of the CBD reactor; and (2) the characterization of the ZnO NPs-treated ND-colored Batik fabric.

#### 2.3.4. Characterization of the ZnO Nanoparticles Collected from the CBD Reactor

To estimate the crystal size of the ZnO nanoparticles, the particles formed during the in situ synthesis were analyzed using X-ray diffraction (XRD) [28,29]. The average particle diameter was evaluated based on the data acquired from the XRD analysis using the Debye–Scherrer equation (Equation (1)) [30].
(1)$$D\;\left(nm\right)=\frac{k\lambda }{\beta cos\theta }$$
where *β* represents the full width at half maximum (FWHM) for the diffraction peak under the consideration (radian), *λ* is the wavelength used in the measurement (0.15406 nm), and *k* is a constant. Furthermore, FESEM and X-ray fluorescence (XRF) were utilized to examine the morphology and Zn content, respectively.

#### 2.3.5. Characterization of ND-Colored Batik Fabric with Immobilized ZnO NPs

The morphology and chemical composition of the ZnO NPs-treated Batik fabrics were characterized using FESEM and EDX, respectively. The antibacterial activity of the ND-colored Batik fabric with immobilized ZnO NPs on its fibers’ surfaces was analyzed against Gram-positive bacteria of *Staphylococcus aureus* using the agar diffusion method. The bacteria were inoculated in a Mueller Hinton Agar medium and placed into petri dishes. Specimens of Batik fabric with a 17 mm radius were placed on the agar surface. The samples were incubated at 37 °C for 24 h. The inhibition zone around the specimen, which is an indication of the antibacterial activity of the Batik fabric [31], was evaluated. The impact of the in situ ZnO NPs immobilization process on the color of the Batik fabric was investigated by a colorimeter (AMT506). The color quality was evaluated using the CIE L*a*b* method, which is a color space that includes all colors that can be seen by the human eye [32]. This color space is a three-dimensional space in three axes, namely L* (brightness), a* (green-red), and b* (blue-yellow). A value of 0 is equal to black, while up to 100 is equal to white. A high L* value means brighter, while the lower L* value means darker. The value of a* leads to red or green, a* positive (+) tends towards red, and a* negative (−) tends towards green. The value of b* leads to yellow or blue. The positive b* (+) tends to be yellow and the negative b* (−) tends to be blue [33]. 

To investigate the influence of ZnO NPs immobilization on the mechanical properties of the Batik fabric, the tensile strength and the elongation of the samples was tested using a Universal Testing Machine (UTM) from JTM Technology (Taiwan). The durability of the ZnO NPs coating on Batik fabrics was evaluated by washing the Batik fabrics according to ISO 105-C06 for 2 laundering cycles. According to this standard, the washing was conducted in 4 g/L detergent at 50 °C for 45 min for each cycle, which is equal to 5 cycles of home laundering. After 2 laundering cycles (equal to 10 cycles of home laundering), the morphology and chemical composition of the samples were evaluated using FESEM and EDX, respectively.

## 3. Results and Discussions

This research used a nitric acid-based leaching solution of EAFD as a precursor for the in situ immobilization of ZnO NPs onto ND (*Jalawe* or *Terminalia bellirica*)-colored fabric. The leaching solution was very acidic, with pH 0, and the EAFD contained a mixture of metals including Zn, Fe, Al, Mn, Ca, Pb, and Sn. To reduce the presence and the influence of metals other than Zn, especially Fe that may give the fabric a brownish red color, the pH of the leaching solution was increased to pH 5 prior to its exposure to the fabric. The in situ immobilization of ZnO NPs was then performed at 50 °C and various pH from 6 to 11. 

### 3.1. Characterization of the Leaching Solution

The metal content in the leaching solution was analyzed using Inductive Coupled Plasma (ICP). Table 1 shows the metal composition in the nitric acid-based leaching solution of EAFD at pH 0 and pH 5.

It is seen in Table 1 that the leaching solution contains several metals, with the largest component being Zn. After precipitation at pH 5, the Zn content was still the greatest at 68,790 ppm, while Fe was reduced significantly. The ratio between Zn and Fe increased significantly from approximately 45 to approximately 105. The low Fe content was intended to avoid the addition of a brownish red color to the fabric. The filtrate resulting from the deposition of pH 5 was then used as a precursor for the in situ synthesis of ZnO on ND-colored Batik fabric. During the in situ immobilization process, particles formed both on the fabric fibers’ surface and in the solution. The particles formed in the solution and then precipitated to the bottom of the reactor. Due to limitations in the characterization of particles that are attached to the fabric’s surface, some characterizations were performed on the particles that precipitated.

### 3.2. Characterization of ZnO Nanoparticles Collected from the CBD Reactor

The in situ immobilization of ZnO onto the ND-colored cotton fabric was carried out using the CBD method in a mild condition of 50 °C and a sufficient base level for the formation of ZnO. To determine the conditions under which ZnO begins to form, pH variations were carried out, at pH 6, 7, 8, 9, 10, and 11. The formed particles were characterized using XRD, as shown in Figure 1. The XRD spectrum of the synthesized particles at pH < 10 (pH 6–pH 9) indicated that they were a lamellar phase compound, namely Pentazinc Octahydroxide Bis nitrate V Dihydrate (Zn_5_(OH)_8_(NO_3_)_2_·2H_2_O). The formation of Pentazinc Octahydroxide compounds is in accordance with the research of Mrad et al. [34] who synthesized ZnO using Zn nitrate and NaOH precursors at 20 °C. Mrad reported that Pentazinc Octahydroxide compounds were formed at pH 10, and by increasing the pH to 13, pure ZnO was formed. Pentazinc Octahydroxide compounds were also reported in the synthesis of ZnO using the precursor Zn nitrate and HMTA using the wet chemical method at 85 °C with an alkaline ratio [OH^−^]/[Zn^2+^] of ½ [35]. Figure 1a indicates that the in situ immobilization process (CBD) at pH 11 could result in the formation of pure ZnO on the fabric’s surface. The peak positions of the XRD spectra obtained at pH 11 appeared at 31.77°, 34.42°, 36.25°, 47.54°, 56.60°, 62.86° and 67.96° corresponding to (100), (002), (101), (102), (110), (103) and (112), respectively. The peaks fit well with the data reported in JCPDS Card No.36–1451 for wurtzite zinc oxide [36], as shown in Figure 1b. 

Based on the results of this study, and in accordance with the study of Mrad et al. [34], the proposed reaction processes for the formation of ZnO and/or lamellar compounds are presented in Equations (2)–(4). The morphology, phases, and surface areas of ZnO nanoparticles depend largely on the amount of cations and anions present in the medium during the preparation [37]. At pH < 10, a reaction (Equation (2)) occurs between cations and anions in the zinc nitrate precursor with NaOH and water. In solution, the number of zinc salt (Zn nitrate) ions were more than the number of OH^−^ ions. It did not yet support the formation of ZnO NPs because it was only formed under strongly basic conditions [34]. The compound formed was Zn_5_(OH)_8_(NO_3_)_2_·2H_2_O according to Equation (2). The formation of these compounds was confirmed by the results of the XRD characterization (Figure 1a, XRD spectra at pH 6, 7, 8, 9). At pH = 10, the number of ions in zinc salt were the same as NaOH ions. Under these conditions, a mixture of compounds, Zn_5_(OH)_8_(NO_3_)_2_·2H_2_O and ZnO NPs, were formed according to Equation (3), confirmed by the results of the XRD characterization. (Figure 1a, XRD spectra at pH = 10). At pH = 11, OH^−^ ions were more abundant in solution, which were causing a strong attraction between the positively charged Zn^2+^ and OH^−^ ions; and afterward, the increased crystallization and formation of ZnO nanoparticles. These conditions correspond to Equation (4) and were shown by the results of the XRD characterization of pure ZnO (Figure 1a, XRD spectra at pH = 11).

pH < 10


(2)
$$\;\left({\mathrm{Zn}}^{2+}+2{\mathrm{NO}}_{3}^{-}\right)+\left({\mathrm{Na}}^{+}+{\mathrm{OH}}^{-}\right)+\frac{1}{4}H_{2\;}\to \;\;\frac{1}{8}{\mathrm{Zn}}_{5\;}{\left(\mathrm{OH}\right)}_{8}{\left({\mathrm{NO}}_{3}\right)}_{2}\cdot 2H_{2}O+\frac{3}{8}\left({\mathrm{Zn}}^{2+}+2{\mathrm{NO}}_{3}^{-}\right)+\left({\mathrm{Na}}^{+}+{\mathrm{NO}}_{3}^{-}\right)$$


2.pH = 10


(3)
$$\;\left({\mathrm{Zn}}^{2+}+2{\mathrm{NO}}_{3}^{-}\right)+2\left({\mathrm{Na}}^{+}+{\mathrm{OH}}^{-}\right)\;\to \;\frac{1}{8}{\mathrm{Zn}}_{5\;}{\left(\mathrm{OH}\right)}_{8}{\left({\mathrm{NO}}_{3}\right)}_{2}\cdot 2H_{2}O+\frac{3}{8}\mathrm{ZnO}+\frac{1}{4}\left({\mathrm{Na}}^{+}+{\mathrm{OH}}^{-}\right)+\frac{7}{4}\left({\mathrm{Na}}^{+}+{\mathrm{NO}}_{3}^{-}\right)+\frac{1}{8}H_{2}O$$


3.pH = 11


(4)
$$\;\left({\mathrm{Zn}}^{2+}+2{\mathrm{NO}}_{3}^{-}\right)+3\;\left({\mathrm{Na}}^{+}+{\mathrm{OH}}^{-}\right)\;\to \;\;\mathrm{ZnO}+2\left({\mathrm{Na}}^{+}+{\mathrm{NO}}_{3}^{-}\right)+\left({\mathrm{Na}}^{+}+{\mathrm{OH}}^{-}\right)+H_{2}O$$


The crystallite sizes of ZnO were estimated with the Scherrer equation (Equation (1)), based on the data obtained from the XRD analysis at pH 11 (Figure 1a). The position and width, peak intensity, and full-width at half-maximum (FWHM) data were identified and used in the estimation of the ZnO particle’s diameter. By using the Scherrer equation (Equation (1)), the ZnO particle’s diameter was estimated to be 26.17 nm, thus, it falls in the category of nanoparticle. The purity of the ZnO nanoparticles, from measurements using XRF, was known to be 98.415%, as shown in Figure 2. The BET Surface area of the ZnO NPs was 9.6692 m^2^/g and the BJH pore size was 14.3211 nm. According to the IUPAC classification, a particle with a pore width of 2 nm to 50 nm is called mesoporous [38]. 

The morphology of the particle formed at various pH is shown by the FESEM image in Figure 3. Figure 3 shows that at pH 6–9, the particle is in the form of a flake, which is a lamellar compound called Pentazinc Octahydroxide Bis nitrate V Dihydrate. The shape of the flake-like material is in accordance with the research of Koao et al. [39], who synthesized ZnO using Zn acetate and ammonia precursors. Figure 3e,f show the transformation of the ZnO particles between pH 10 and pH 11. At pH 11 (Figure 3f) pure ZnO formed in the shape of a nanorod. Based on the analyzed XRD and FESEM data, we hypothesized that, in our experimental range, the most suitable conditions for the in situ formation of ZnO NPs on ND-colored fabric would be at 50 °C and at pH 11 in a CBD. 

### 3.3. Characterization of ND-Colored Batik Fabric with Immobilized ZnO NPs

To compare the morphology of the particles in the solution and on the fabric’s surface, FESEM characterization was then carried out on the treated fabric at various pH, and the resulting images are shown in Figure 4. It was assumed that the chemical structure of the particles attached to the fabric’s surface are identical to the ones in the solution. Figure 4 shows a clear difference between the attached Pentazinc Octahydroxide (Figure 4a–e), i.e., flake-like particles, and the attached ZnO NPs (Figure 4f) on the fabric’s fiber surface. Interestingly, the ZnO NPs particles attached to the fiber did not show the same morphology as those in the solution (nanorods). This demonstrates that the ZnO NPs nucleate (and grow) on the fibers rather than forming in solution and subsequently adsorb onto the fibers (heterogenous nucleation). In addition, Aladpoosh and Montazer [40] suggested that the entrapment of ZnO seeds between the cellulosic chains of cotton limits the growth of ZnO NPs, leading to the altered morphology.

The interaction between the ZnO NPs and the cellulosic chains of cotton fabric (Batik fabric) was analyzed using FT-IR. Figure 5 demonstrates the FT-IR spectra of untreated (Figure 5a), ZnO NPs-treated ND-colored Batik fabric (Figure 5b), and ZnO NPs powder (Figure 5c). The bands of untreated fabric appear at 3410 cm^−1^ (O-H stretching), 2900 cm^−1^ (C-H stretching) [40], 1635 cm^−1^ (C=O stretching), 1427 cm^−1^ (C-H wagging), 1319 cm^−1^ (C-H bending) [22], and 1026 cm^−1^ (C-O stretching vibration) [41]. Compared to the spectra of untreated fabric, Figure 5b shows that the band at 2900 cm^−1^ shifted to 2916 cm^−1^, indicating that C-H was affected by other chemical bonds. The band at 1026 cm^−1^ shifted to 1033 cm^−1^, indicating that C-O was also affected by other chemical bonds. Figure 5b shows bands below 500 cm^−1^, which can be attributed to Zn-O bonds [22,41], as indicated by Figure 5c.

The mechanism for the synthesis of ZnO NP in situ on the cellulose chain of cotton fabrics is divided into three main stages: pre-nucleation, nucleation, and growth [40]. The pre-nucleation stage occurs when the cellulose of cotton cloth is stirred in a Zn salt solution. Zn^2+^ ions will be dispersed in the solution. ND-colored cellulose cotton fibers produce a slightly negative charge on the surface due to the ionization of hydroxyl groups when immersed in water. Zinc ions with a positive charge can absorb into the cotton fibers due to coordination bonding, forming a Zinc–Cellulose complex [42]. Similarly, Zinc ions can also form coordination bonding with the ND molecules that are attached to the cellulose fibers, forming a Zinc–ND complex (Figure 6). It is known from the literature [43] that the active compound of *Jalawe* extract is ellagic acid. Furthermore, heterogeneous nucleation theory stipulates that the nucleation energy barrier is lowered on a surface if attractive forces exist between the surface and (at least one type of) the building blocks [44]. The ND-colored cotton surface attracts Zn^2+^ ions, which, in turn, attract the OH^−^ ions and form the nucleus for ZnO crystals [42], as shown in Figure 6.

It is worth noting, however, that if the interactions between Zn^2+^ and the ND-colored-cotton fabric are too strong, this may hinder nucleation, as observed by Palms et al. [44]. When the interactions are too strong, Zn^2+^ ions become immobile and are unable to coalesce in large enough quantities, or in the required spatial configuration, to form a nucleus. This condition, known as “frustration” in crystallography, occurs when external factors arrange the building blocks in a way that is incompatible with the final crystal structure, preventing the formation of nuclei of critical size. It can be seen, from Figure 4, that this was not the case for our system, since the ZnO crystals form in similar numbers all-over the fiber’s surface. Nucleation occurs when enough NaOH is added into the solution to surpass the critical nucleation concentration. Additionally, alkaline conditions will change the non-active ND-colored-cellulose (Cell-OH) into active ND-colored-cellulose (Cell-O-) to form a cellulose complex with Zn^2+^ ions (zinc-cellulose complex). The ZnO crystal growth stage occurs due to excess sodium hydroxide in solution. The creation of an alkaline environment causes swelling of the cotton fabric and destroys the intermolecular hydrogen bonds that facilitate particle penetration into the fiber’s structure. 

Based on the above-described discussions and mechanism, it is believed that the interactions between the ZnO NP and the ND-colored cotton fabric are van der Waals and/or coordination bonding. 

### 3.4. Antibacterial Activity of ND-Colored Batik Fabric with Immobilized ZnO NPs

The antibacterial activity assessment of the ZnO treated natural Batik fabric samples was conducted using a *Staphylococcus aureus* on agar diffusion technique. *S aureus* is the most frequently evaluated species, commonly found in the human skin, nose, respiratory tract, and in home laundry processes [45]. The bacteria is the leading cause of skin and soft tissue infections, such as abscesses, cellulitis, and furuncles. *S aureus* can cause serious infections, such as pneumonia, bloodstream infections, and joint infections [46]. 

The antibacterial assessment was conducted for all treated Batik fabrics and untreated Batik fabric was used as a control. The antibacterial properties were shown by an inhibition zone around the samples. Table 2 shows the inhibition zone for all Batik fabric samples against *S aureus* bacteria. There was no inhibition zone on untreated Batik and treated Batik fabric at pH 6–pH 9, as shown in Figure 7. The inhibition zone was visible on treated Batik fabric at pH 10 and pH 11, as shown in Figure 8. Treatment at pH 6–pH 9 does not create antibacterial properties, this is in accordance with the study by Ghosh et al. [47], who found that cotton fabrics exhibit antibacterial properties at pH < 6, and there is no inhibition zone at pH ≥ 6. This indicates that the antibacterial activities of treated Batik fabric at pH 10 and pH 11 in this study are due to the presence of ZnO NPs. 

Figure 8 shows that the untreated Batik fabric specimen (Figure 8a) attracted bacterial growth. On the other hand, both of the specimens with immobilized ZnO NPs (Figure 8b,c) showed an inhibition zone around the specimens. Furthermore, the inhibition zone of the treated Batik fabric at pH 11 is slightly greater than that of the treated Batik fabric at pH 10, which we believe to be due to the pure ZnO NPs content in the treated Batik at pH 11. The antibacterial mechanism of ZnO NPs described by researchers is caused by the formation of Reactive Oxygen Species (ROS) that can damage bacterial cell membranes [48,49,50]. Many researchers reported the effect of the shape and size of ZnO NPs to their antibacterial activities [48,51,52]. Smaller sizes of ZnO NPs can easily penetrate into bacterial cell membranes due to their large interfacial area, thus enhancing their antibacterial activities. Nanorod nanoparticles showed enhanced antibacterial properties against pathogenic bacteria compared to the hexagonal ZnO NPs [52]. Xu et al. [51] reported that sharp-edged ZnO NPs may have greater antibacterial activities because they more easily penetrate bacterial cell membranes than nanoparticles with smooth edges.

### 3.5. The Influence of the In Situ Immobilization Process of ZnO NPs on Color Quality

The influence of the ZnO NPs in situ immobilization process at various pH on the color quality of the ND-colored fabric was evaluated using organoleptic (visual) observation and the CIE L*a*b* method, as shown in Figure 9 and Table 3, respectively. 

It can be seen in Figure 9 that dyeing using *Jalawe* extract resulted in a yellowish color direction, which agreed with the b* positive value being greater than the a* value (Table 3). The L* value of treated Batik at all pH was greater than that of untreated samples, indicating that the application of ZnO on ND-colored Batik produced brighter color when compared to the untreated one. This phenomenon might be due to two reasons. First, the interaction between the cations and anions presents in the bath solution. The in situ immobilization process involved the ND-colored cellulose fabric with Al^3+^ mordant that was immersed in bath solutions at various pH 6–11. The leaching solutions contain cations (as showed in Table 1, such as Zn^2+^, Ca^2+^, Fe^2+^) and anions such as (NO_3_)^−^, and OH^−^ with various concentrations, depending on the immobilization pH. Similarly to the mordanting ions, the cations can react with the cellulose and ND molecules, while the anions can create bonding with the mordanting ion. Both processes may lead to the leaching of the ND from the cellulose fiber, and thus color change of the ND-colored fabric to brighter colors. Second, because of the photocatalytic properties of ZnO NPs. When ZnO NPs absorb radiation energy, the electrons are excited to the conduction band, leaving behind holes in the valence band. In the presence of water and oxygen, oxidation occurs in which photo-induced positive holes are involved, and reduction, in which photo-induced negative electrons are involved. In these reactions, highly reactive oxygen species (ROS) are formed that can react with ND molecules causing degradation. Saeed et al. [53] proposed that the degradation of alizarin red in aqueous media was caused by the photocatalytic properties of ZnO imparted on polymers. However, based on a study by Reningtyas et al. [54], who applied ZnO NPs to indigo-dyed cotton fabrics, they found that the photocatalyst properties of ZnO NPs were not dominant. The author proposed that the reason for this was that the lack of water in the cotton’s surface prevented the formation of OH radicals. Our study is similar to Reningtyas’, therefore, we propose that the reason for the color change in the ZnO NPs-treated Batik fabric is because of the interaction between cations and anions present in the bath solution (first reason). 

The color change, as indicated by the CIE L*a*b method, agreed with the organoleptic observation (Figure 9) in which, visually, there is a gradient of color from untreated to treated fabric across the varied pH range, from pH 6–11, i.e., it becomes brighter and brighter towards a higher pH. These results are consistent with previous studies [30,51], which stated that the application of ZnO NPs produced a brighter color. However, based on organoleptic observations, the color of the Batik fabric treated at pH 11 is still acceptable for *Jalawe* ND-colored Batik fabric.

### 3.6. The Influence of the In Situ Immobilization Process of ZnO NPs on the Mechanical Properties

Table 4 shows the mechanical properties of untreated and ZnO NPs-treated Batik fabric. In situ immobilization of ZnO NPs on the Batik fabric enhanced its tensile strength in both the warp and weft directions. This is in agreement with previous studies [23,40,55,56]. This might be due to the creation of bonding between the cellulosic chain and ZnO NPs. Moreover, ZnO NPs acted as a filler between the polymeric chains, which increased the tensile strength of the fabric [40]. On the other hand, ZnO NPs’ incorporation onto the Batik fabric reduced the elongation in both the warp and weft directions. This is possibly because the application of ZnO NPs led to a more rigid and stiffer fabric than the untreated sample, thereby reducing the fabric’s elasticity [23,25,55]. 

### 3.7. The Durability of ZnO NPs Immobilization on ND-Colored Batik Fabrics

The morphology and chemical composition of Batik fabrics before and after two laundering cycles are shown in Figure 10. It can be seen that after two washing cycles (equal to 10 cycles of home laundering), a significant amount of ZnO NPs remained on the Batik fabrics, although some of the agglomerated nanoparticles leached. The EDX analysis in Figure 10c shows the chemical composition of ZnO NPs-treated Batik fabric before washing. The elemental spectrum shows the presence of carbon (C), oxygen (O), zinc (Zn) and aluminum (Al). The C and O elements are the constituent materials of cellulose (cotton fabric). The Zn element confirmed the presence of ZnO on the surface of the treated Batik fabrics. The Al element indicated the alum mordant of the ND-colored Batik fabrics. Importantly, the EDX analysis confirmed there was no Pb element on the ZnO NPs-treated Batik fabric. The Pb element, a heavy metal that is dangerous to humans, was detected in the leaching solution (Table 1) in a small concentration. Thus, the CBD and ZnO NPs formation process performed in this study could exclude this element from the final product.

From the EDX analysis (Figure 10c,d), it can be seen, quantitatively, that the concentration of ZnO NPs before washing was 5.92% and after washing it was reduced to 5.10%, which corresponds to the removal of 16% of the ZnO NPs. Thus, 84% of the ZnO NPs remained on the Batik fabrics after washing, demonstrating a sufficient durability of their attachment to the fabrics [56]. In our previous study, we found that the durability of the antibacterial properties of ZnO NPs in in situ treated Batik fabrics was equivalent to more than 35 cycles of home laundering [26].

## 4. Conclusions

We have successfully immobilized ZnO NPs on natural dye (ND)-colored Batik fabric at low temperature using an in situ CBD method. Electric Arc Furnace Dust was utilized as the source of the ZnO NPs. The process was performed at 50 °C and various pH from 6 to 11. At pH 6–pH 9, the particle formed is Pentazinc Octahydroxide Bis nitrate V Dihydrate (Zn_5_(OH)_8_(NO_3_)_2_·2H_2_O). At pH 10, a phase transition from Pentazinc Octahydroxide Bis nitrate V Dihydrate to ZnO NPs nanorods was observed. The formation of pure ZnO NPs was observed at pH 11 and, based on FESEM characterization, it was imparted to the fabric. It was assumed that the interactions between the ZnO NPs and ND-colored Batik fabric were van der Waals or coordination bonding. The assesment of durability revealed that after two washing cycles (equal to 10 cycles of home laundering), 84% of the ZnO NPs remained on the Batik fabrics. In situ immobilization of ZnO NPs onto Batik fabric enhanced its tensile strength and reduced its elongation. Based on antibacterial characterization using the agar diffusion method, the Batik fabric treated at pH 10 and pH 11 had antibacterial properties against *Staphylococcus aureus*. Characterization using the CIE L*a*b* scale showed that the immobilization process affects the color quality of the ND-colored Batik fabric, i.e., higher pH results in brighter color. Leaching of some ND molecules due to interactions with cations and anions in the bath solution was believed to be responsible for the observed phenomenon. From a fundamental point of view, this study provides insights into the surface science and technology of natural dye-coloured cellulosic fabric. From an application point of view, the fundamental knowledge of surface science sheds light on important parameters for the preparation of sustainable antibacterial Batik fabric products. In addition, an added value was created for Electric Arc Furnace Dust (EAFD) waste, which was used as source of the ZnO NPs.

## Figures and Tables

**Figure 1 polymers-15-00746-f001:**
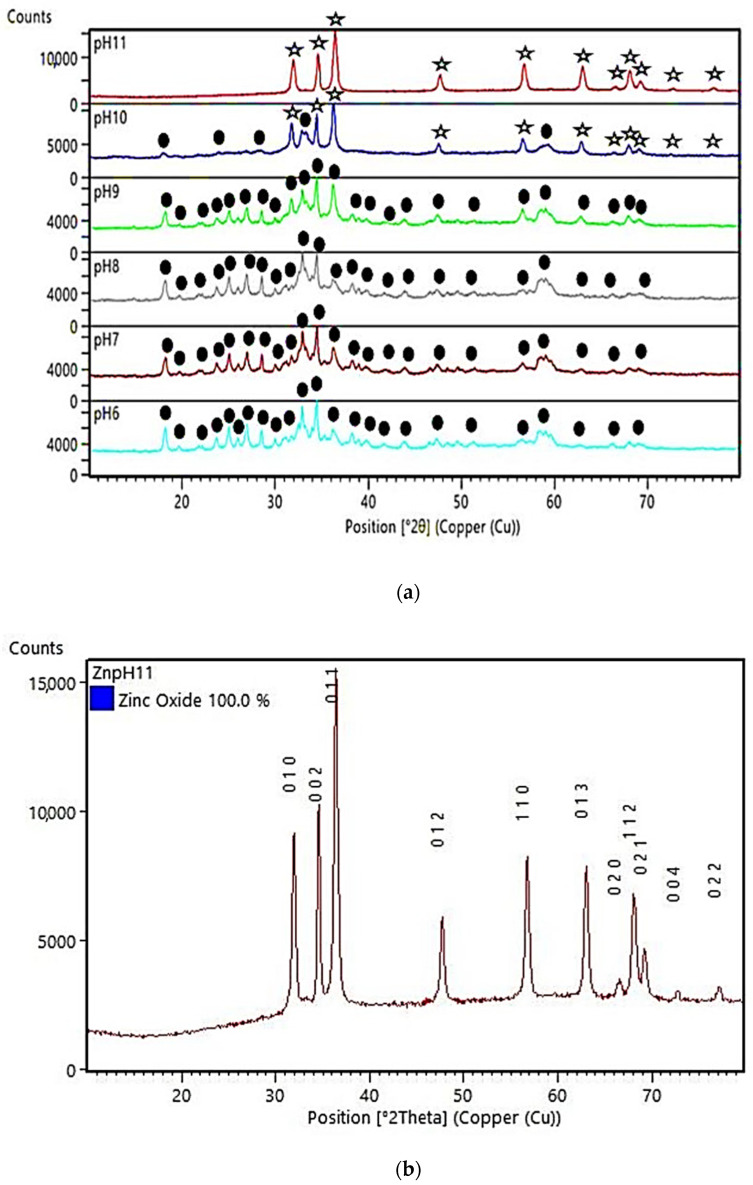
(**a**) XRD spectra of particles formed in the in situ synthesis of ZnO on ND-colored Batik fabric at pH 6–pH 11, 
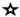
: ZnO, 
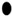
: Zn_5_(OH)_8_(NO_3_)_2_·2H_2_O (Pentazinc Octahydroxide), (**b**) XRD spectra of JCPDS data.

**Figure 2 polymers-15-00746-f002:**
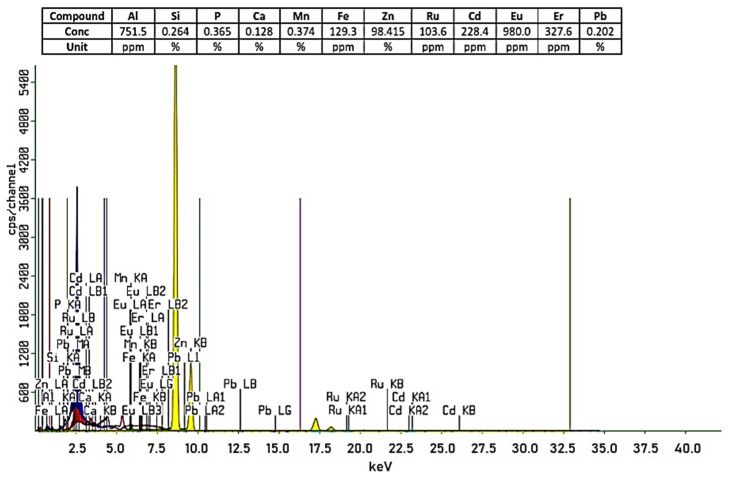
Chemical composition of ZnO from XRF analysis.

**Figure 3 polymers-15-00746-f003:**
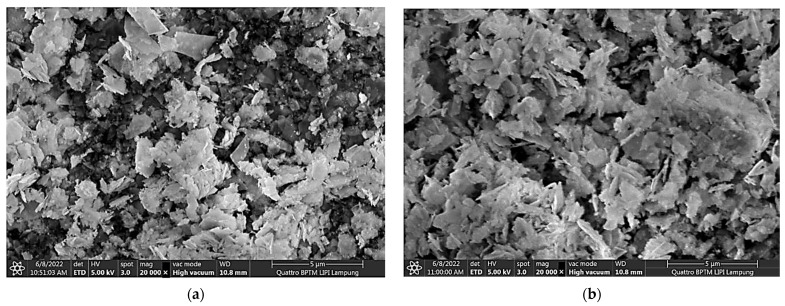
Morphology (FESEM image) of the particle formed in the in situ synthesis of ZnO NPs on ND-colored Batik fabrics at (**a**) pH 6, (**b**) pH 7, (**c**) pH 8, (**d**) pH 9, (**e**) pH 10, (**f**) pH 11.

**Figure 4 polymers-15-00746-f004:**
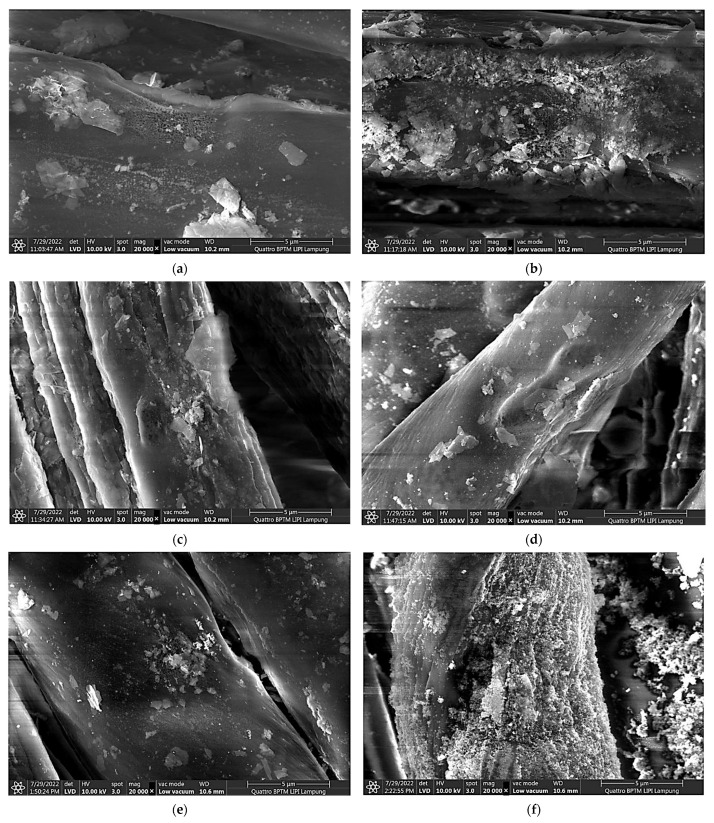
Morphology (FESEM image) of ND-colored Batik fabric surface after in situ immobilization of ZnO NPs at various pH: (**a**) pH 6, (**b**) pH 7, (**c**) pH 8, (**d**) pH 9, (**e**) pH 10, (**f**) pH 11.

**Figure 5 polymers-15-00746-f005:**
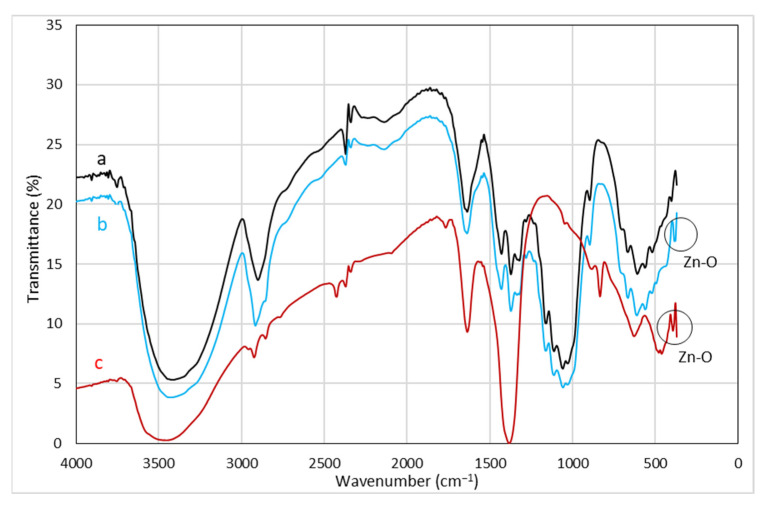
FT-IR spectra of (a) untreated fabric, (b) ZnO NPs-treated ND-colored Batik fabric, (c) ZnO NPs powder.

**Figure 6 polymers-15-00746-f006:**
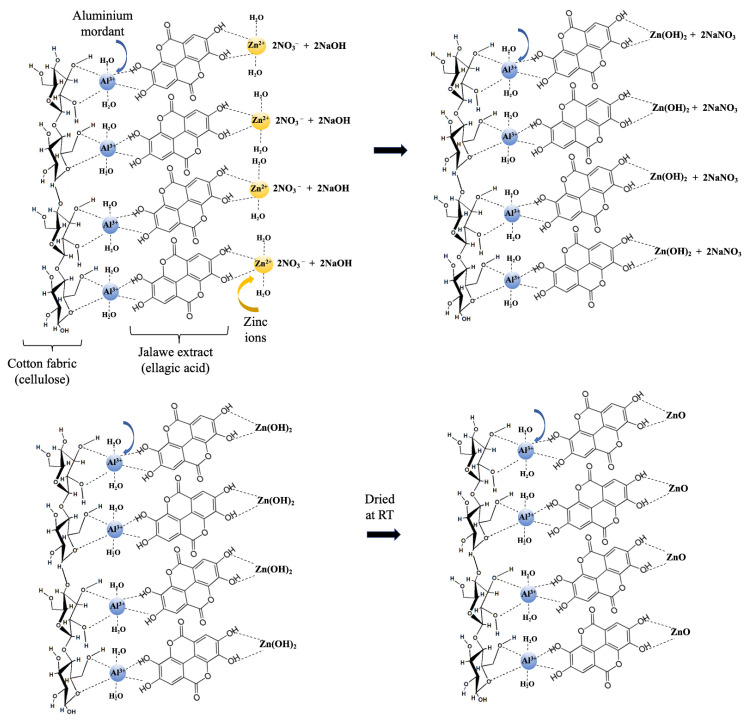
The chemical reaction pathways for the formation of ZnO NPs on ND-colored cotton fabric [42].

**Figure 7 polymers-15-00746-f007:**
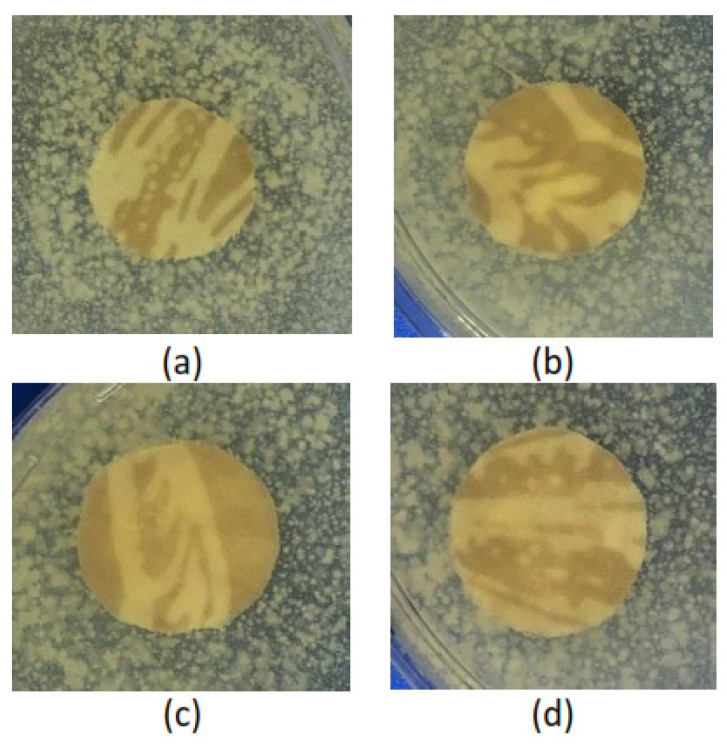
Photograph of the antibacterial assessment (no inhibition zone) of (**a**) treated Batik fabric at pH 6, (**b**) treated Batik fabric at pH 7, (**c**) treated Batik fabric at pH 8, (**d**) treated Batik fabric at pH 9.

**Figure 8 polymers-15-00746-f008:**
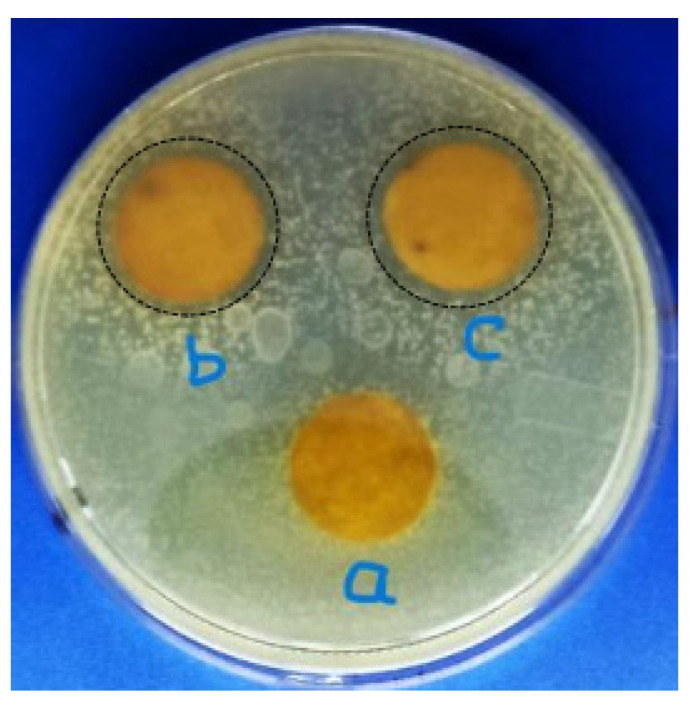
Photograph of the antibacterial activities of (**a**) untreated Batik fabric, (**b**) treated Batik fabric at pH 10, (**c**) treated Batik fabric at pH 11. The black dashed circles are guides for the eye and represent inhibition zones. The diameter of the samples are 17 mm.

**Figure 9 polymers-15-00746-f009:**
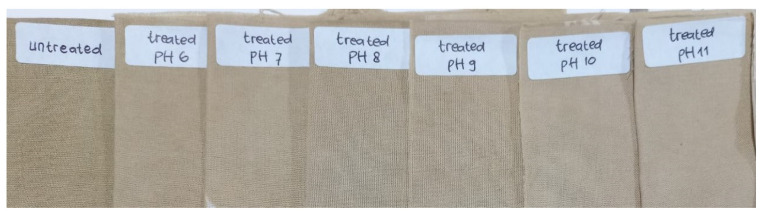
The effect of the in situ ZnO NPs immobilization process at various pH on the color quality of ND-colored Batik fabric.

**Figure 10 polymers-15-00746-f010:**
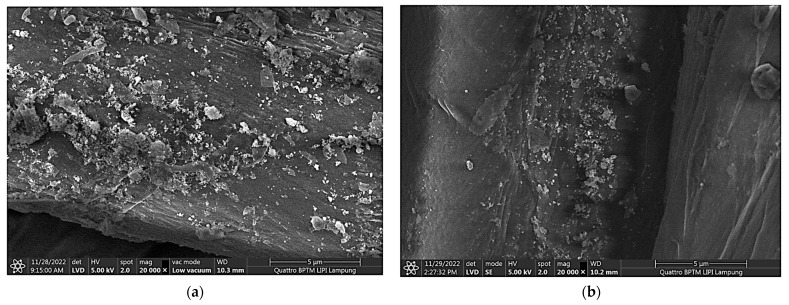
(**a**,**b**) Morphology (FESEM image) of the ND-colored Batik fabric surface before and after washing, respectively. (**c**,**d**) Chemical composition of ND-colored Batik fabric from the EDX analysis before and after washing, respectively.

**Table 1 polymers-15-00746-t001:** Metal composition in the leaching solution of EAFD with nitric acid 3 M using ICP analysis.

Metal Composition	Concentration (ppm)	
	pH 0	pH 5
Zn	122,300	68,790
Fe	2735	656
Al	100	7
Mn	230	72
Ca	1166	1101
Pb	1198	143
Sn	3461	828

**Table 2 polymers-15-00746-t002:** The Inhibition Zone of the Batik Fabric Samples.

Batik Fabric Samples	Diameter of Inhibition Zones (mm)
Untreated Batik fabric	0
Treated Batik at pH 6	0
Treated Batik at pH 7	0
Treated Batik at pH 8	0
Treated Batik at pH 9	0
Treated Batik at pH 10	22.72 ± 0.50
Treated Batik at pH 11	23.13 ± 0.25

**Table 3 polymers-15-00746-t003:** The value of L*, a*, and b* of untreated and ZnO NPs-treated ND-colored Batik fabric.

Batik Fabric Samples	L*	a*	b*
Untreated Batik fabric	65.86	6.37	21.53
Treated Batik at pH 6	67.88	7.00	21.02
Treated Batik at pH 7	68.06	6.84	21.47
Treated Batik at pH 8	69.75	6.48	19.32
Treated Batik at pH 9	69.92	7.12	20.68
Treated Batik at pH 10	70.00	6.78	19.74
Treated Batik at pH 11	70.30	7.00	20.13

L*: brightness, a*: green-red, b*: blue-yellow.

**Table 4 polymers-15-00746-t004:** The mechanical properties of untreated and treated Batik fabric.

Samples	Tensile Strength (N)		Elongation (mm)	
	Warp	Weft	Warp	Weft
Untreated Batik fabric	121.89 ± 2.26	67.50 ± 1.38	23.00 ± 1.66	33.25 ± 1.08
Treated Batik at pH 11	123.85 ± 4.01	67.93 ± 3.91	15.50 ± 0.43	28.25 ± 1.08

## Data Availability

Not applicable.

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
