# Peer review of "Low Temperature In Situ Synthesis of ZnO Nanoparticles from Electric Arc Furnace Dust (EAFD) Waste to Impart Antibacterial Properties on Natural Dye-Colored Batik Fabrics"

_polymers, 2023, doi:10.3390/polym15030746_

Round 1

Reviewer 1 Report

Comments to the authors

The manuscript reports interesting results concerning the In Situ Synthesis of ZnO Nanoparticles from Electric Arc Furnace Dust for imparting antibacterial properties to the natural dye-colored Batik fabrics. Development of textiles with functional properties have received increased attention in the last decades. The present work affords a promising strategy for the designing Batik fabric with functional properties. I suggest minor revision for quality enhancement, which are given bellow.

1.      The manuscript contains grammatical and typing mistakes which should be carefully read and remove and the language need improvement.

2.      Discuss some catalytic using and citing this article Applied Water Science (2021) 11:105 https://doi.org/10.1007/s13201-021-01442-0

3.      Discuss some biological applications of ZnO NPs.

4.      In section 2.2, there is no need to mention stove, table, basin, glassware etc. Remove these names and keep only characterization instruments.

5.      Page 5, line 198, what does pH 0 means?

6.      Page 5, table 1, correct the “68.790” as “68,790”.

7.      There is no graph provided for XRF.

8.      The authors should properly correlate the XRD data with the results presented in the manuscript.

9.      Write ZnO NPs instead of ZnONP throughout the whole manuscript.

10.  Discus the obtained scientific results in the conclusion section.

Author Response

Dear Reviewer,

Thank you for the suggestions, we have improved our manuscript according to all suggestions.

Reviewer 2 Report

The manuscript, “Low Temperature In Situ Synthesis of ZnO Nanoparticles from Electric Arc Furnace Dust (EAFD) Waste to Impart Antibacterial Properties on Natural Dye-Colored Batik Fabrics” reports the development of in-situ ZnO NPs immobilized natural dye-colored batik fabrics and their anti-microbial properties. The reported study is interesting and results are promising. However, manuscript should be improved in certain aspects as given in the comments. Therefore, the manuscript can be accepted after a moderate revision.

1. Provide the JCPDS data in Fig. 1

2. The formation mechanism of pure ZnO at pH 11 should be discussed

3. How about the leaching of ZnO NPs from the fabrics upon washing?

4. Provide the BET surface area analysis of the samples

5. How about the ZnO NPs dosage-dependent antimicrobial properties of fabric?

6. What kind of interaction exists between the ZnO NPs and fabrics? Support with some relevant characterizations

7. How about the of effect of pH on the antimicrobial activity of the fabric irrespective of presence of ZnO?

8. Why it is tried only with Staphylococcus aureus, why not other species?

9. Any insights into other properties of fabrics such as mechanical property after the immobilization of ZnO NPs?

10. English of the manuscript should be improved

Author Response

Dear Reviewer,

Thank you for the suggestions. We have improved our manuscript according to all suggestions.

Reviewer 3 Report

In this article, the authors studied utilizing one type of industrial waste to introduce ZnO nanoparticles and how different conditions affect the color of the fabric product. The results are interesting and the materials characterization is impressive. However, there are some concerns about the design of the experiment and several important typos in the manuscript. It is suggested that the authors to carefully proofread the manuscript before publication. Detailed comments are given below.

1.       Although using industry waste could be economically beneficial, the waste may contain unknown toxic compounds that is harmful to the human health and environment. For example, it contains lead, which is very dangerous to human. This fabric materials might be used to contact human skins directly. It is suggested for the authors to carefully discuss these concerns.

2.       Line 129, the authors wrote, “a dust grain size of <75 m (passed a 200 mesh sieve).” It is concerning that the authors are using 75 m dusts. Also, the meaning of “200  mesh” is unclear.

3.       Line 16, the full name of “UNESCO” should be explained when used for the first time.

Author Response

Dear Reviewer,

Thank you for the suggestions. We have already improved our manuscript according to all suggestions

Round 2

Reviewer 2 Report

Authors have revised the manuscript satisfactorily and it can be accepted for the publication. 

Author Response

Dear Reviewer

Thank you very much for the suggestion
